# A Multicenter, Investigator-Blinded, Randomized Controlled Trial to Assess the Efficacy of Calf Neuromuscular Electrical Stimulation Program on Walking Performance in Peripheral Artery Disease: The ELECTRO-PAD Study Protocol

**DOI:** 10.3390/jcm11247261

**Published:** 2022-12-07

**Authors:** Alexis Le Faucheur, Pierre Jéhannin, Adrien Chanteau, Pauline Blanc-Petitjean, Alan Donnelly, Clément Hoffmann, Samir Henni, Alessandra Bura-Rivière, Adrien Kaladji, Damien Lanéelle, Guillaume Mahé

**Affiliations:** 1Clinical Investigation Center, INSERM, CIC 1414, F-35033 Rennes, France; 2Univ. Rennes, M2S—EA 7470, F-35000 Rennes, France; 3Univ. Rennes, F-35000 Rennes, France; 4Epidemiology and Public Health Unit, CHU Rennes, F-35033 Rennes, France; 5PESS Department, Health Research Institute, University of Limerick, V94 T9PX Limerick, Ireland; 6Vascular Medicine Unit, CHU Brest, F-29200 Brest, France; 7Vascular Medicine Department, CHU Angers, F-49000 Angers, France; 8UMR CNRS 6015, INSERM 1083, Univ. Angers, F-49000 Angers, France; 9Vascular Medicine Unit, CHU Toulouse, F-31000 Toulouse, France; 10Vascular Surgery Unit, CHU Rennes, University Hospital, F-35033 Rennes, France; 11Vascular Medicine Unit, CHU Caen-Normandie, University Hospital, F-14033 Caen, France; 12Vascular Medicine Unit, CHU Rennes, University Hospital, F-35033 Rennes, France

**Keywords:** electric stimulation therapy, intermittent claudication, walking capacity, muscle function

## Abstract

This paper describes a currently on-going multicenter, randomized controlled trial designed to assess the efficacy of calf neuromuscular electrical stimulation (NMES) on changes in maximal walking distance in people with lower extremity peripheral artery disease (PAD), compared with a non-intervention control-group. This study (NCT03795103) encompasses five participating centers in France. PAD participants with a predominant claudication at the calf level and a maximal treadmill walking distance ≤300 m are randomized into one of the two groups: NMES group or Control group. The NMES program consists of a 12-week program of electrical stimulations at the calf-muscle level. The primary outcome of the study is the change in maximal treadmill walking distance at 12 weeks. Main secondary outcomes include changes in the pain-free treadmill walking distance; 6 min total walking distance; global positioning system (GPS)-measured outdoor walking capacity; daily physical activity level by accelerometry; self-reported walking impairment; self-reported quality of life; ankle-brachial index; and skin microvascular function, both at the forearm and calf levels. Recruitment started in September 2019 and data collection is expected to end in November 2022.

## 1. Introduction

Lower extremity peripheral artery disease (PAD) is a serious public health concern with an estimated prevalence in people aged 25 years and older of 5.5% worldwide (236.62 million people), 7.99% in the European Region (51.1 million people) and 9.79% in France (4.42 million people) [1]. PAD is associated with an increased risk of disability and mortality as compared to people without PAD [2]. Atherosclerosis is the main cause of PAD and causes the chronic narrowing of arteries that induces lower limb blood-flow impairment [3,4]. This hemodynamic impairment can induce ischemic pain which severely impairs the performance of daily physical activities such as walking [5], which progressively becomes very disabling and isolates people living with PAD [6].

Current care for PAD patients includes lifestyle modification (i.e., cessation of smoking and dietary habit modifications) and medical therapy to lower the cardiovascular risk [7]. In addition, before considering a possible revascularization procedure (surgical or endovascular), supervised or home-base exercise should be proposed as the first-line therapy to decrease leg symptoms and improve walking capacity [3,8]. Compared to medical treatment alone, interventions such as supervised exercise have resulted in enhancements in walking capacity [8] due to an improvement in endothelial vasodilator function or skeletal-muscle adaptations [9]. However, there is a lack of easy access to structured exercise therapy due to several barriers resulting in this efficacious therapy remaining largely unavailable to the majority of patients [10]. Furthermore, PAD patients with walking pain symptoms are often unable or reluctant to partake in exercise [10,11].

Given the need for additional therapy strategies, additional methods such as neuromuscular electrical stimulation (NMES) have been proposed [12,13,14]. Interestingly, Abraham et al. [15] showed that NMES of calf muscles results in a significant increase of arterial inflow without measurable muscle ischemia or pain. Several vascular and/or nonvascular pathways have been proposed to support the potential efficacy of NMES on the improvement of walking performance in people with PAD [13]. However, a recent systematic review concluded that it was not yet possible to draw a conclusion on the potential efficacy of NMES on walking performance in people with PAD because of the very low number of available studies with adequate sample size and low risk of bias [13]. Moreover, the two available randomized controlled trials (RCTs) testing home-based NMES programs alone to improve walking performance included short-duration NMES programs (1 month) [12,14]. Maximal walking distance (MWD) was increased in these two studies (by 39 m to 40 m, or 34% to 35%). However, it is likely that a longer duration intervention of 3 months, a more usual duration in exercise-therapy interventions, would have induced larger improvements in walking performance.

The primary objective of the ELECTRO-PAD study is to assess the efficacy of NMES on changes in MWD after a home-based 12-week program in people with PAD by comparing two groups: (1) an intervention group receiving the NMES program, and (2) a non-intervention control group with no intervention. Both groups receive standard care including usual medical treatment and usual non-structured walking incitation to walk on their own daily.

## 2. Materials and Methods

### 2.1. Study Design

The ELECTRO-PAD study is a prospective, randomized, multicenter, single-blind (investigator-blinded), superiority clinical trial with two parallel groups and a primary endpoint of MWD change at 12 weeks following an NMES program. Recruitment for the study commenced in September 2019 and the study end is planned for November 2022.

The study protocol is reported here in accordance with the SPIRIT 2013 recommendations for reporting standard protocol items for Clinical Trials [16]. Information required for those recommendations which could not be directly reported in the manuscript are presented in Appendix A. Figure 1 shows the schedule of pre-screening, enrolment, interventions, and assessments for the ELECTRO-PAD study. The protocol was reviewed and approved by the sponsor and the applicable French institutional ethical committee (CPP: “*Comité de Protection des Personnes*”) with respect to scientific content and compliance with applicable research and human subjects’ regulations, according to the French law and the Declaration of Helsinki (CPP reference 2018/07). The ELECTRO-PAD study was registered on ClinicalTrials.gov (NCT03795103). A Clinical Research Officer mandated by the promoter ensured participant safety and good practices of the clinical study. The coordinating center was certified to conduct such randomized controlled trial.

In parallel to this clinical trial, two ancillary studies were performed. The first ancillary study was performed only in PAD participants included in the Rennes coordinating center due to material availability. The aim of this ancillary study was to assess the skin microvascular function in the included participants both at visits #1 and #2, between 1 and 14 days after the corresponding visit (Figure 1). A second ancillary study was also performed, but on a distinct group of healthy participants with no intervention. The main aim of this second ancillary study was to determine the normal range for most of the different outcome measures that were performed in PAD participants. In that way, procedures performed by this group of healthy participants were almost the same as those performed by the participants with PAD. Inclusion and Exclusion criteria for healthy participants are reported in Appendix A (item 2b).

### 2.2. Recruitment, Randomization, and Blinding

Participants are being identified and recruited by the investigators from five French university hospitals located in the northwest (n = 4) and the southwest (n = 1) of France: Rennes (coordinating center), Angers, Brest, Caen, and Toulouse. In each center, participants are screened for eligibility during their usual medical appointment for PAD management, and during which a modified Strandness treadmill walking test (3.2 km/h, 10%) was performed to assess walking ability (see outcomes assessment). Eligible participants who meet the inclusion criteria (Table 1) were asked to participate in the study. A letter presenting the objectives and the protocol sequence is given to the participant (Appendix A) and, after ≥ a 48-h period, the participant is called by phone by a member of the local staff. If the participant agrees to participate, the inclusion visit (visit #1) is planned, preferably within the next 7 to 21 days (Figure 1). This strategy for participant enrolment is designed to: (i) minimize the burden of planning numerous inclusion visits (visit #1) while the potentially eligible participants had in fact a MWD > 300 m (exclusion criteria); and (ii) test the reproducibility of the treadmill walking test by assessing the variability of the MWD [17]. No inclusion percentage to achieve was given to each center. The enrollment period is anticipated to last ~43 months from study commencement (including enrollment of healthy participants in the second ancillary study).

Inclusion and exclusion criteria for participants with PAD are displayed in Table 1. In addition, criteria for discontinuing allocated interventions are presented in Appendix A (Item 11b). Participants provide paper-based written, informed consent before they participate in the study (Appendix A), which is collected by the assessor at visit #1 (inclusion visit, Figure 1). At visit #1, the order of the treadmill test and 6 min test is first randomized electronically, and the information is given to the assessor of the local center where the participant performs both tests accordingly. Once visit #1 is achieved, following an electronic process (Appendix A, Item 16a), each participant is randomized into one of the two arms (1:1) of the study: (1) NMES group; or (2) Control group (CONT). Randomization is performed as block randomization with a 1:1 allocation. The randomization is stratified and equilibrated (block for equilibration not provided here to preserve concealment allocation) in the five inclusion centers. Only the biostatistician, the data manager, and the research associate at the coordinating center have access to the allocation sequence to preserve the allocation concealment. In that way, the assessor of the different outcomes in each center is blinded to the participant treatment group allocation. The research associate of the coordinating center receives an email mentioning the participant group allocation. Then the research associate mails a box containing the following to the participant’s home:for both groups of participants: (i) a box containing the monitors for daily physical activity (PA) and outdoor walking-capacity assessment; (ii) a handbook that explains how to wear the monitors, how to perform the outdoor walking sessions, and that includes a diary to report different information (see below and Appendix A); (iii) a sealed envelope containing a brochure with advice for regular PA on a day-to-day basis (Appendix A).for the NMES group only: (i) an opaque box containing the NMES device; (ii) a sealed envelope containing the NMES handbook for implementation and follow-up of the NMES program (Appendix A).

At the coordinating center, the box is directly hand-delivered to the participant. The research associate calls on each participant once the box is received at their home (identified via a parcel-tracking system). In that way, the research associate indicates the appropriate box to open first and that contains PA monitors. The research associate then explains how to install and wear the monitors verbally while the participants read the guidebook that contains all the information needed. A second call phone is scheduled at the end of the 7-day free-living PA-measurement period during which the research associate discloses the group allocation to the participant.

Due to the nature of the intervention and the follow-up, the participants and the research associate cannot be blinded to allocation, but participants are strongly encouraged not to disclose their allocation status during visit #2. Due to personnel availability at the coordinating center, the assessor of skin microvascular function assessment at both visits (ancillary study #1), and the 6 min walking tests at visit #2 is not blinded to the participant’s treatment arm.

### 2.3. NMES Intervention

Participants allocated in the NMES group are equipped with a VEINOPLUS^®^ device (Ad Rem Technology, Paris, France). The device consists of a handheld central unit that works on a battery, connected to two ovoid skin-adhesive electrodes (Figure 2). Each electrode is ~5 cm wide and 10 cm long. For the participants with mainly unilateral symptoms limiting walking performance, the two electrodes are positioned on the symptomatic and limiting calf (Figure 2). The stimulation consists of a series of rectangular pulses of low energy (<25 mC), low voltage (50 Vpeak), and low frequency (1–250 Hz), with maximum duration of impulse of 240 ms. Intensity can be set manually. The shape of the current wave is biphasic, leading to nearly symmetric contractions of the heads of the calf muscle. For the participants with bilateral calf symptoms, two electrodes are positioned first on their right or left calf during half of the program session, then on the other calf for the other half of the session. Participants are asked to adopt a sitting or semi-extended position for better stimulation efficiency. The participant can perform other intellectual activities during the session but should avoid movements and displacements. The intensity of stimulation is regulated by the participants themselves because there is interindividual variability in the felt effect of electrical stimulation [15]. The intensity is indicated on the screen of the device by a number corresponding to the level of intensity reached (Figure 2). The participants are asked to select gradually the highest possible stimulation intensity until he/she sees and feels deep contractions of the calf without pain.

The NMES program is performed at home and is delivered 5 days a week across the program (i.e., 12 weeks) with 2 h/day of NMES on weeks 3 to 6 and weeks 9 to 12, and 3 h/day on weeks 1, 2, 7, and 8. Each NMES session lasts a minimum of 1 h and a maximum of 2 h. The time of the day to complete the sessions is left free to the participants (Appendix A).

To enhance validity of data, for each NMES session, participants are asked to report the following information in a diary: stimulated calf(s), NMES intensity level maintained during the session, cumulated total duration reached at the end of the session, and feelings about the session. To assess protocol adherence, the ratio of the total duration of NMES reached at the end of the protocol over the expected total duration will be used in future analyses. In addition, phone calls are also conducted every two weeks (±4 days) to record information about the NMES program to ensure the correct functioning of the device, check the correct number of hours of NMES performed, promote participant retention, and complete follow-up. The participants are also asked to indicate whether any changes in symptoms, comorbidities or walking activity potentially occurred. Furthermore, participants in the NMES group receive a brochure with advice for engaging in daily PA (Appendix A).

### 2.4. Non-Intervention Control Group

Participants in the CONT group only receive a brochure with advice for engaging in daily PA. Phone calls are also conducted every two weeks (±4 days) to record any potential symptoms, comorbidities, or walking activity changes, promote participant retention, and complete follow-up.

### 2.5. Primary and Secondary Outcomes

Details are presented in Appendix A (Item 2b). The primary outcome of the ELECTRO-PAD study will be the change in MWD at 12 weeks as assessed by a modified Strandness treadmill test (see outcomes assessment).

Secondary outcomes will be changes at 12 weeks in: (i) treadmill pain-free walking distance (PFWD); (ii) 6 min total walking distance; (iii) global positioning system (GPS)-measured outdoor walking capacity; (iv) ankle-brachial index (ABI); (v) daily PA level, including walking pain-manifestation assessment (see below); (vi) walking impairment questionnaire (WIQ) sub-scales and total scores; (vii) short-form health survey (SF36) score; (viii) peripheral artery questionnaire (PAQ) score; (ix) dietary questionnaire score; (x) delta from resting oxygen pressure (DROP) using TcPO_2_ during treadmill walking test and (xi) skin microvascular function assessed both at the forearm and calf levels using the laser speckle contrast imaging (LSCI) technique; and (xii) maximal walking distance according to the location of the arterial obstruction (using scan images).

### 2.6. Outcomes Assessment (Visits #1 and #2)

#### 2.6.1. Primary Outcome

*Modified Strandness treadmill walking test.* During this test, the walking speed is held constant (2 mph, i.e., 3.2 km/h) and the slope is fixed at 10% [18]. The participant is encouraged to walk for the longest time possible. The treadmill walking test is discontinued at the participant’s request, or by protocol, up to a maximum exercise duration of 20 min. A scale of pain (0 = no pain; 1 = onset of pain; 2 = moderate pain; 3 = intense pain; 4 = maximal pain) is used to quantify the appearance and development of pain during the walking test as well as pain relief during recovery [19]. Participants indicate the start of their claudication pain, from which the PFWD is computed, and the test stops at the point when the participant does not want to continue owing to lower limb pain, from which the MWD is computed. A 12-lead electrocardiogram monitors heart rate and cardiac electrical activity during the exercise-test procedure. In the Rennes center alone (due to equipment availability), gas exchange is assessed using a metabolic cart (MasterScreen CPX [MSCPX], Jaeger^®^, Germany). The system is calibrated according to the manufacturer’s instructions before each treadmill test.

To our knowledge, the minimal clinically important difference in the Strandness MWD has not been defined among patients with PAD. In MWD measurements obtained from a standardized graded maximal treadmill test (Gardner-Skinner walking test) among patients with PAD [20], a small clinically meaningful change was defined as ~107 m (i.e., 121 s) while a large clinically meaningful change was defined as ~214 m (i.e., 241 s).

#### 2.6.2. Secondary Outcomes

*Questionnaires.* The walking impairment questionnaire (WIQ) is used to assess the degree of functional impairment reported by the participants for different walking tasks. The WIQ is a valid tool to detect improvement or deterioration in the daily walking ability of participants with intermittent claudication [21,22]. The higher the WIQ score, the better the participant’s functional walking ability. The quality of life of the participants is assessed by the SF36 and by the Peripheral Artery Questionnaire (PAQ). The SF36 includes 36 questions that yields an eight-scale score of well-being and functional health, and two psychometrically based physical and mental health summary measures [23]. Each scale is directly transformed into a 0–100 scale on the assumption that each question carries equal weight. The lower the score, the greater the disability. The higher the score, the lesser the disability, i.e., a score of zero is equivalent to maximum disability and a score of 100 is equivalent to no disability [23]. The PAQ is a 20-item questionnaire that quantifies participants’ physical limitations, symptoms, social function, treatment satisfaction, and quality of life [24]. Its clinical validity has been confirmed in 711 patients with PAD [25]. Finally, dietary habits of participants are evaluated by a validated food frequency questionnaire to assess the diet associated with atherosclerotic diseases [26].

*Ankle-brachial index.* Measurement of ABI is performed according to American Heart Association recommendations [27] using a hand-held Doppler probe (8 MHz; Basic Atys Medical, Soucieu en Jarrest, France).

*Exercise transcutaneous oxygen pressure measurement (Exercise TcPO_2_).* This measure is performed only in equipped centers and aims to estimate the degree of ischemia during the modified Strandness treadmill walking test [7,28,29]. TcPO_2_ assessment detects regional blood-flow impairment at the proximal and distal limb simultaneously and bilaterally during exercise. The measurements from the TcPO_2_ electrodes allow the calculation of the DROP index (expressed in mmHg) and the absolute change in TcPO_2_ from resting value in each of the four limb electrodes, corrected for the absolute change in TcPO_2_ at the chest electrode [28,30].

*6 min walking test.* This test has been widely used in clinical trials in people with PAD to assess walking performance [8,31] and is performed according to standard procedure [31,32]. Among patients with PAD, following supervised and home-based exercise programs (pooled data), a small clinically meaningful change was defined as 12 m and a large clinically meaningful change was defined as 34 m [20]. From the study of the test–retest reliability of the 6 min walking test, Sandberg et al. [33] reported a minimal detectable change of 46 m.

*Daily PA level and walking pain experience.* Following both visits, the daily PA level of PAD participants is assessed over a consecutive 7-day period. We use a combination of PA monitors to progress in the knowledge of PA monitoring in PAD. Following both visits, during a consecutive period of seven days (excluding a preliminary day for habituation), participants are asked to wear a waist belt (right hip) holding two monitors: an accelerometer wGT3X (wGT3X+/wGT3X-BT models; Actigraph, Shalimar, FL, USA) and a GPS receiver (Qstarz BT-Q1000XT (Qstarz International Co., Ltd., Taipei, Taiwan). In addition, participants wear an accelerometer ActivPAL™ (ActivPAL4, PAL Technologies©, Glasgow, Scotland) adhered to the middle of the right thigh with a fixing system (Tegarderm^®^). A StepWatch pedometer is also worn at the right ankle level. Initially, participants wore the StepWatch 3.0 (OrthocareInnovations, Oklahoma City, OK, USA) but its non-rechargeable and non-replaceable battery had a limited life span. Thus, participants are now wearing the new 4.0 model (Modus Health™ LLC, Edmonds, WA, USA) during the protocol. The participants are carefully instructed to wear the monitors throughout the day for seven consecutive days and to remove the monitors during water-based activities (e.g., showering and swimming) and at bedtime; the exception is the activPAL4 that can be worn during showers and during the night. All the instructions for wearing the monitors are summarized in a handbook (Appendix A) that is sent to the participants together with the monitors. Participants are asked to report their bedtime and waking time every day in the handbook. In addition, at the end of each day, the participants report the main indoor and/or outdoor activities carried out throughout the day with a timestamp (Appendix A). Data from accelerometers will be processed following previously published procedures [34,35]. The main metrics computed to assess daily PA level will be the following: mean time per day of total PA; mean time per day of light, moderate, and vigorous PA; mean number of steps per day; mean number of steps per day (or mean daily stepping time) accumulated in different duration bouts; and mean number of steps per day (or mean daily stepping time) accumulated in different cadence bands.

Finally, to objectively quantify the pain experienced by participants during daily life walking, participants wear a watch on the wrist (right or left) and are asked to press event marker button(s) to indicate events related to walking-pain manifestations and stops induced by walking pain (Appendix A). The procedure has been previously described and validated [36]. Initially, the Micro Motionlogger^®^ watch (WatchAmbulatory Monitoring, Inc., Ardsley, NY; firmware Action-W version 2) was used at the beginning of the protocol but was progressively replaced by a new and more ergonomic home-based App developed on Android™ and deployed on Wear OS smartwatches (Huawei Watch GT or Fossil Carlyle, Appendix A). During the 7-day free-living PA measurement, an SMS is sent every morning to remind participants to wear the PA monitors. After the 7-day PA monitoring, participants send back monitors and the PA guide by mail using a prepaid package.

*GPS-measured outdoor walking capacity.* During the 7-day free-living PA measurement, participants are asked to complete two 30 min outdoor walking sessions while wearing and using the PA monitors as described above. The instructions to perform the walk were previously reported [37,38,39] and are summarized in the handbook (Appendix A). Data from the GPS receiver will be directly analyzed to process indicators of walking capacity, as previously reported [36,37,38,39]. For each participant, the main GPS parameters recorded will be: the total distance walked over the outdoor session; the whole duration of the outdoor session; the number of walking and stopping bouts; the mean walking speed calculated over each walking bout; the total time and distance over each walking bout; the total time over each stopping bout; the bout with the highest time and distance walked; and the mean time and distance over all the walking bouts. By combining the information recorded on the watch and GPS data, the outdoor PFWT and MWT will be computed [36]. Since the outdoor walking capacity depends both on walking speed and grade, the metabolic equivalent of task per min (MET·min) will be computed for each walking bout using previously published predictive equations for walking metabolic rate [40,41].

*Skin microvascular function assessment.* This measure is also performed only on participants included in Rennes due to device availability. Skin microcirculation is measured with the LSCI to assess the modifications of endothelial function at calf and forearm levels [42]. The skin microvascular function is measured after a fasting period. Two different tests are performed: The post-ischemic reactive hyperemia (PORH) and the local thermal hyperemia tests. The PORH test assesses the increase in skin blood flow to tissue that follows the release of a brief 3 min arterial occlusion [43]. The cuff is inflated at a supra-systolic pressure (50 mmHg above the systolic pressure). The PORH test, which is expected to be endothelium dependent and involving both myogenic and metabolic factors, evaluates the reperfusion of the vascular beds and is characterized by a peak in the skin blood flow after the cuff release [42]. Local thermal hyperemia is obtained with a special probe (42 °C–44 °C) that leads to a sustained increase in skin blood flow that is biphasic: an initial peak that is axon-reflex mediated, and a sustained plateau phase that depends on nitric oxide [44]. The LSCI was preferred to the laser Doppler flowmetry due to a better reproducibility [45]. Skin blood flow is expressed in arbitrary units or as multiples of the baseline.

### 2.7. Sample Size

The calculation of the number of participants per group was based on the expected improvement in the MWD after 12 weeks in the intervention group (NMES) compared with the control group. Considering previous studies [12,14] that assessed the effects of NMES during a program of lower duration (1 month), we expected a 50% improvement in MWD in the NMES group with no significant change in the CONT group. We assumed a baseline MWD of ~115 ± 78 m in both groups. Thus, considering a between-groups difference of approximately 58 m with a standard deviation of 78 m at the end of the 12-week program, the minimum number of participants to be included in each group is approximately 29, with an alpha risk of 0.05 and a power of 80%. We anticipated that the total number of participants to be included could be 80 (40 per group), allowing for a maximum ~30% dropout rate.

### 2.8. Data Collection, Management, and Analysis

*Data collection and management.* The data collected are in an electronic Case Report Form (eCRF) and data monitoring is carried out on a regular basis (~every six months).

*Statistical analysis (see also Appendix A, item 20a).* Descriptive statistics will be performed to describe the baseline characteristics of interest at the level of the two groups and for the different outcomes. Continuous data will be expressed as mean +/− standard deviation or median values with interquartile range, depending on the distribution. A normality test will be carried out to judge the Gaussian or non-Gaussian distribution of the data of the continuous variables studied. Two-sample, two-sided t-tests (or Mann–Whitney U test) and χ2 tests will be used to compare continuous and categorical characteristics of participants across the two groups at baseline, respectively.

For statistical analysis of the primary and secondary outcomes after study completion, two -sample, two-sided t-tests (or Mann–Whitney U test if the data is not normally distributed) will be used to compare changes in outcomes between baseline and the 3-month follow-up between the NMES and the CONT groups. However, in case of imbalance in baseline characteristics between the 2 groups, outcome data will be analyzed using linear regression models adjusting for significantly different baseline data between the two groups [46].

Firstly, an intention-to-treat analysis will be performed. As recommended [47], a missing at random (MAR) assumption will be made for missing data for principal analysis by performing complete cases and multiple imputation analysis. Deviation from MAR will also be assessed with sensitivity analysis under the missing not at random (MNAR) assumption, using controlled multiple imputation with different controlled scenarios. A per-protocol analysis will be also performed considering only participants with data available both at visit #1 and visit #2.

A subgroup analysis will be performed to explore the potential association between change in treadmill MWD and reported walking activity changes during the follow-up. In the NMES group, potential association between the total dose of NMES over the program and change in treadmill MWD will be assessed.

Statistical analysis will be carried out using R software [48]. The level of significance will be fixed at *p* < 0.05.

## 3. Discussion

Lower-extremity electrical stimulation is an older technique that has regained recent interest in the treatment of lower extremity functional impairment of people with PAD, as shown by recently published original and protocol studies [49,50,51,52]. Owing to the very low number of available studies with adequate sample size and low risk of bias, a recent systematic review concluded that no clear clinical indication could be drawn regarding the efficacy of NMES for the management of impaired walking function in patients with PAD [13]. The multicenter, investigator-blinded, randomized controlled ELECTRO-PAD clinical trial will contribute to increase the level of scientifical evidence regarding the efficacy on NMES on walking capacity in people with lower extremity PAD.

One strength of the ELECTRO-PAD clinical trial is the duration of the NMES program. As compared to previous studies that assessed the effect of an NMES program on walking capacity during four-week or a six-week programs [12,14,49], the NMES program of the ELECTRO-PAD trial is longer, with a three-month program duration which duplicates the recommended duration for SET programs [8]. It will be of interest to observe the magnitude of change (if any) in walking capacity from the ELECTRO-PAD trial compared to previous NMES programs and SET programs.

Another strength of the ELECTRO-PAD clinical trial relies on the methods used for the assessment of walking capacity. Compared with previous studies [12,14,49], the ELECTRO-PAD clinical trial will assess the walking capacity of PAD participants through three complementary walking assessment tests: the modified Strandness treadmill test, the six-minute walking test, and a GPS-assessment of outdoor walking capacity. Although this increases the complexity of the protocol, this methodologic choice was deemed important considering the existing controversy about the best functional test for measuring response to interventions in PAD [31,53,54]. It has been shown that walking capacity assessment tests provide outcomes that are not interchangeable in participants with PAD [55]. The ELECTRO-PAD clinical trial will show whether, and how, an NMES program impacts differently walking capacity changes according to the outcome measure that is considered.

Potential study limitations should also be considered. First, in the design of the ELECTRO-PAD clinical trial, careful consideration was given to the definition of the control group since the lack of an appropriate control group was clearly a major bias in previous studies [13]. For instance, the two only available randomized controlled trials testing home-based NMES programs alone to improve walking performance used transcutaneous electrical nerve stimulation as a placebo [12,14], whereas this type of lower-extremity electrical stimulation was reported to increase walking capacity in PAD participants in another study [56]. In the ELECTRO-PAD study, the choice was made to use a control group with participants receiving lifestyle advice, including walking advice, together with best medical therapy, as this was considered to be the best solution both at the methodological (intention to treat approach) and ethical levels [13]. Indeed, although the use of a sham intervention could be seen as an ideal way to design double-blind studies, it raises issues with regards to the ethics of volunteer deception and the risk of volunteer awareness of the sham treatment, with a subsequent risk of unbalanced dropout rates between groups. In addition, following a 12-week program would exacerbate the sedentary behavior and worsen the functional impairment of a sham control group [57].

Second, supervised exercise therapy (SEP) is recommended as the first-line therapy to decrease leg symptoms and improve walking capacity in people with PAD [3,8]; thus, administering SEP to the control group could have been considered as an attractive option. When designing the ELECTRO-PAD clinical trial, the choice was made not to use a SEP-intervention control group since the availability of SEP programs in France is extremely low. Even when available, SEP programs are shorter in duration (~3–4 weeks) compared to current practice guidelines [8]. Undoubtedly, this would have been seen as a major limitation when comparing walking capacity changes between the control (SEP) group and the NMES group. Interestingly, Babber et al. recently showed a significant adjunctive benefit of a footplate NMES program in PFWD improvement, but not in MWD improvement, when NMES was used as an adjunct to SEP compared to SEP alone [49].

Finally, the participants with bilateral lesions apply NMES on one calf during half of the program session, then on the other calf for the other half-session. Thus, the duration of the NMES session for each limb for participants with bilateral lesions is two times less than the session duration for participants with a unilateral lesion. The device used does not allow the user to simultaneously connect two pairs of electrodes (i.e., one for each limb). Another possible solution could have been to use two devices simultaneously in participants with bilateral lesions. However, we did not choose this solution because we expected this could have been too burdensome for those participants, increasing the risk of low adherence to the NMES program or dropout. The inclusion of participants with bilateral lesions is deemed important for a wider applicability of NMES intervention in people with PAD in the future. We foresee that there will be few participants with walking limitations due to strictly equal bilateral symptoms at the calf level. Nevertheless, we will assess whether the change in maximal walking distance differs according to the participant’s symptom profile (i.e., bilateral vs unilateral).

## Figures and Tables

**Figure 1 jcm-11-07261-f001:**
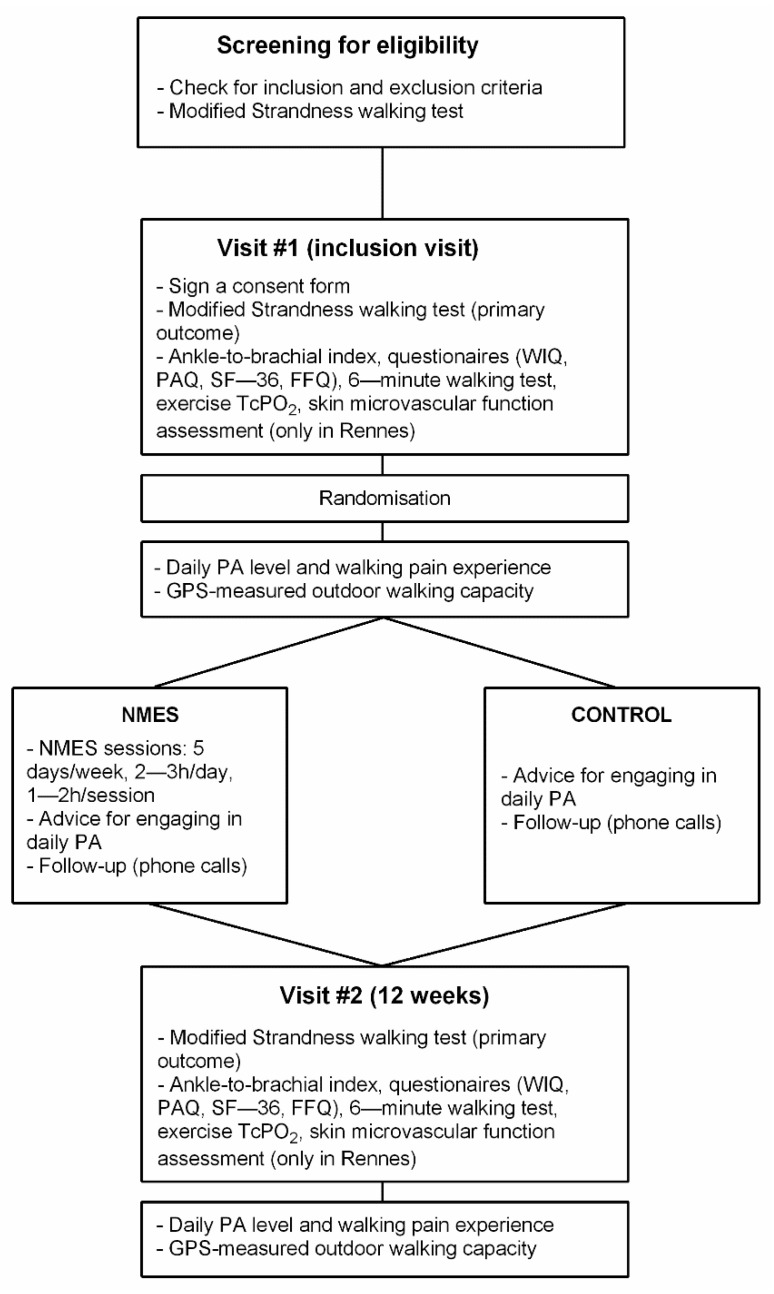
Flowchart for participant screening for eligibility, inclusion, assessments, randomization, interventions, and follow-up. ***Legend.*** WIQ: Walking Impairment Questionnaire; PAQ: Peripheral Artery Questionnaire; FFQ: Food Frequency Questionnaire; TcPO_2_: Transcutaneous oxygen pressure measurement; PA: Physical Activity; GPS: Global Positioning System; NMES: Neuromuscular Electrical Stimulation. Note that participants’ allocation to NNMS or CONTROL groups following randomization is concealed to the participants until they complete the 7-day physical activity measurement.

**Figure 2 jcm-11-07261-f002:**
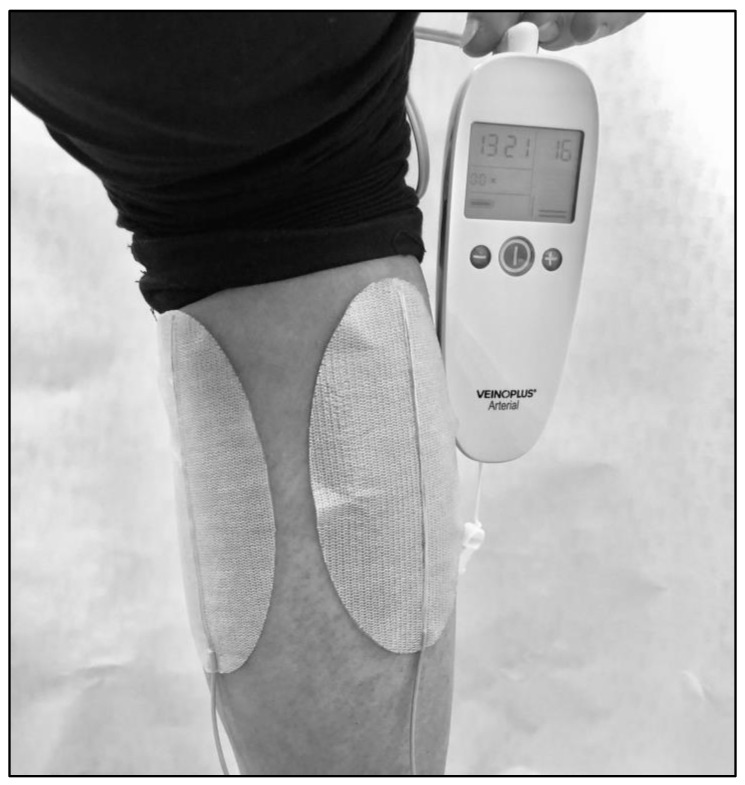
Photo showing the VEINOPLUS^®^ device at the calf level.

**Table 1 jcm-11-07261-t001:** Inclusion and exclusion criteria of the ELECTRO-PAD clinical trial.

**Inclusion Criteria**
Age > 40 years old. Subjects with Lower Extremity Peripheral Artery disease (LEPAD). LEPAD is defined by the presence of at least one of the following criteria: History of revascularization in the lower limbs due to LEPAD OR Ankle brachial index (ABI) of ≤0.90 ^1^ OR ABI or ankle systolic blood pressure decrease during recovery from treadmill walking test >20% or >30 mmHg, respectively OR Toe-brachial index ≤ 0.70 if ABI cannot be measured and if incompressible arteries are suspected Complain of exertional calf pain (fatigue, discomfort or cramping) that can begin or not at rest, causes the participant to stop walking and relieves or lessens within 10 min of rest (assessed using the San Diego questionnaire AND confirmed during treadmill testing). Pain (fatigue, discomfort, or cramping) that is mainly located at the calf level. Maximal walking distance on treadmill < 300 m (treadmill protocol: 3.2 km/h, 10% grade). Subject receiving for at least one month the recommended medical therapy for LEPAD management (antiplatelet therapy and statin medication). Obtained informed consent.
**Exclusion Criteria**
Patients with a pacemaker or defibrillator. Patients with acute or critical limb ischemia. Ambulation limited by exertional symptoms other than intermittent claudication (e.g., dyspnea or angina pectoris). Ambulation limited by exertional symptoms indicative of intermittent claudication but affecting muscles in the lower extremities other than the calves. Contraindication to exercise testing according to the American Heart Association and the American College of Sports Medicine. Major cardiovascular event (myocardial infarction or stroke) or major surgery within the previous three months before inclusion. Female patients who are pregnant, planning to become pregnant, or lactating. Known presence of an aneurysm of the abdominal aorta > 4 cm or an aneurysm of the iliac artery > 1.5 cm. Patient subject to legal protection (guardianship or tutelage measure) and persons deprived of their liberty (according to French law). Simultaneous participation in another ongoing clinical research protocol. Unwilling or unable to engage in the completion of a 12-week program. Any planned event(s) that could interfere with the completion of the protocol: e.g., extended holidays preventing the completion of the intervention or planned hospitalization for a prolonged period. Body mass > 160 kg (may exceed treadmill limit). Inability to understand and sign informed consent forms due to cognitive or language barriers. LEPAD due to causes other than atherosclerosis.

^1^ PAD is defined by an ankle–brachial index (ABI; the ratio of the systolic blood pressure at the ankle to the systolic blood pressure in the arm) of 0.90 or less. An ABI is considered as “borderline” between 0.91 and 0.99, “normal” between 1.00 and 1.40, or “noncompressible” when >1.40 [3].

## Data Availability

Not applicable.

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
