# Peer review of "A Multicenter, Investigator-Blinded, Randomized Controlled Trial to Assess the Efficacy of Calf Neuromuscular Electrical Stimulation Program on Walking Performance in Peripheral Artery Disease: The ELECTRO-PAD Study Protocol"

_jcm, 2022, doi:10.3390/jcm11247261_

Round 1

Reviewer 1 Report

The manuscript presents the protocol of a currently on-going multicenter RCT to assess the efficacy of calf neuromuscular electrical stimulation in PAD. The study appears methodologically correct and well described. Therefore, I have no further comments to add.

Author Response

We thank the Reviewer #1 for his/her reviewing and we are pleased to hear that the study protocol appears methodologically correct and well described.

Reviewer 2 Report

I read with interest the article by Le Faucheur et al "A multicenter, investigator-blinded, randomized controlled trial to assess the efficacy of calf neuromuscular electrical stimulation program on walking performance in peripheral artery disease: The ELECTRO-PAD study protocol". This manuscript presents the protocol of an ongoing study to investigate the possibility of using NMES in the rehabilitation of patients with PAD. Compared with previous studies, the authors increase the training time (from 1 to 3 months), while they hope to get more meaningful scientific results. This multicenter study is carefully designed and registered with ClinicalTrials.gov (NCT03795103), which also indicates the possibility of obtaining meaningful scientific results. Nevertheless, when reviewing the manuscript, I had questions about which I would like to receive clarification from the authors.

1. The study protocol was started in September 2019 and should end in November 2022. Why did the authors decide to publish the manuscript with the protocol after 3 years of the study, when it is already ending and it is time to publish the results? Perhaps there were problems with the recruitment of patients in the study? I would like an explanation.

2. The authors chose maximal walking distance in the treadmill test as the primary endpoint, while the most widely used six-minute walk test was chosen as the secondary endpoint. Why not vice versa?

3. Somewhat surprising is the number of secondary endpoints (the authors suggest estimating eleven such endpoints). When choosing a large number of endpoints, problems arise in the further statistical processing of the material. Multiple endpoints increase sample size compared to a single endpoint, as sufficient power must be provided to test all hypotheses. Complicated and the calculation of the required sample size in multiple endpoints. However, the authors only calculate the required sample size for a single endpoint (maximum walking distance in the treadmill test). Then a natural question arises - is there a need to evaluate such a number of secondary endpoints if it is not possible to obtain correct statistical results?

4. The authors plan in case of imbalance in baseline characteristics between the 2 groups, outcome data will be analyzed using linear regression models adjusting for the baseline data significantly different between the two groups. Are these linear regression models also planned for all 11 secondary points? In addition, some of these parameters will be evaluated on a smaller sample of patients (at the Rennes Clinic).

5. About the NMES protocol. Did I understand correctly that with a bilateral lesion, the duration of the NMES session for each limb will be two times less than with a unilateral lesion? What is the reason for the different duration of NMES per day in different weeks of the program (2 hours or 3 hours)?

6. Patients with PAD are characterized by low motivation to participate in physical rehabilitation programs. How do the authors of the study plan to solve this problem? Wouldn't it turn out that the NMES group will have the most motivated patients for rehabilitation, which will lead to a bias in the results?

Author Response

We sincerely thank the Reviewer #2 for the quality of his/her reviewing and his/her relevant comments. A detailed response is provided to each comment, below.

  1. The study protocol was started in September 2019 and should end in November 2022. Why did the authors decide to publish the manuscript with the protocol after 3 years of the study, when it is already ending and it is time to publish the results? Perhaps there were problems with the recruitment of patients in the study? I would like an explanation.

We understand that the Reviewer #2 is wondering why this manuscript is submitted for publication only at the end of the study.

As suggested by the Reviewer #2, the recruitment of patients has been longer as expected. There were two main reasons:

1/ Since the treatment assessed could only be applied to the calves’ level, we only included participants with PAD for who the lower limb(s) pain was mainly located at the calves' level, thereby inducing walking limitation. Thus, due to this quite strict criterion, a significant number of patients that were screened could not be included because they could also report pain at the buttock, thigh and/or foot levels that could also potentially limit them during walking.

2/ Due to the COVID pandemic, we had to repeatedly stop the inclusion visits.

Consequently, considering these reasons as well as the modest size of our research team, we put all our efforts in including patients and we had few times to finalize the writing of the protocol. That’s the only reason.

  1. The authors chose maximal walking distance in the treadmill test as the primary endpoint, while the most widely used six-minute walk test was chosen as the secondary endpoint. Why not vice versa?

We thank the Reviewer #2 for this interesting comment.

As reminded in the discussion section of the manuscript, at the time of the conception of the ELECTRO-PAD protocol there was an existing controversy about the best functional test for measuring response to interventions in PAD (McDermott el al., 2014; Le Faucheur et al., 2015; Hiatt et al., 2014). In the 2016 AHA/ACC Guideline on the Management of Patients With Lower Extremity Peripheral Artery Disease, exercise treadmill is recommended as a useful method to objectively assess functional status, whereas the six-minute walk test is not mentioned (Gerhard-Herman et al., 2017). More recently, in the 2019 AHA SCIENTIFIC STATEMENT on Optimal Exercise Programs for Patients With Peripheral Artery Disease, both the treadmill test and the 6-MWT are presented as the two main outcome measures used to assess changes in walking endurance and peak exercise capacity in response to exercise interventions in patients with PAD. However, no indication is given about the superiority of one test over the other.

We agree with Reviewer #2 that the 6-MWT is widely and more and more used since a decade to assess walking capacity change to interventions in people with PAD. However, as reminded by Treat-Jacobson et al. (2019), treadmill exercise performance has been the most common outcome measurement used. That’s one of the main reasons that explains why at the time of the conception of the ELECTRO-PAD protocol, the choice was made to use the maximal walking distance in the treadmill test as the primary endpoint.

The Reviewer #2’ comment is relevant since it has been recently shown that walking capacity assessment tests provide outcomes that are not interchangeable in participants with PAD (McDermott, 2020). Interestingly, as underlined in the discussion section of the original version of the manuscript, the ELECTRO-PAD clinical trial will assess the walking capacity of PAD participants through three complementary walking assessment tests: the modified Strandness treadmill test, the six minutes walking test and a GPS-assessment of outdoor walking capacity. Although a secondary endpoint, this will provide interesting exploratory results about how a NMES program could impact differently walking capacity changes according to the outcome measure that is considered.

References

McDermott, M.M.; Guralnik, J.M.; Criqui, M.H.; Liu, K.; Kibbe, M.R.; Ferrucci, L. Six-minute walk is a better outcome measure than treadmill walking tests in therapeutic trials of patients with peripheral artery disease. Circulation 2014, 130, 61-68, doi:10.1161/CIRCULATIONAHA.114.007002.

McDermott, M.M.; Guralnik, J.M.; Tian, L.; Zhao, L.; Polonsky, T.S.; Kibbe, M.R.; Criqui, M.H.; Zhang, D.; Conte, M.S.; Domanchuk, K.; et al. Comparing 6-minute walk versus treadmill walking distance as outcomes in randomized trials of peripheral artery disease. J Vasc Surg 2020, 71, 988-1001, doi:10.1016/j.jvs.2019.05.058.

Le Faucheur, A.; de Mullenheim, P.Y.; Mahe, G. Letter by Le Faucheur et al regarding articles, "Six-minute walk is a better outcome measure than treadmill walking tests in therapeutic trials of patients with peripheral artery disease" and "The treadmill is a better functional test than the 6-minute walk test in therapeutic trials of patients with peripheral artery disease". Circulation 2015, 131, e406, doi:10.1161/CIRCULATIONAHA.114.012389.

Hiatt, W.R.; Rogers, R.K.; Brass, E.P. The treadmill is a better functional test than the 6-minute walk test in therapeutic trials of patients with peripheral artery disease. Circulation 2014, 130, 69-78, doi:10.1161/CIRCULATIONAHA.113.007003.

Gerhard-Herman, M.D.; Gornik, H.L.; Barrett, C.; Barshes, N.R.; Corriere, M.A.; Drachman, D.E.; Fleisher, L.A.; Fowkes, F.G.; Hamburg, N.M.; Kinlay, S.; et al. 2016 AHA/ACC Guideline on the Management of Patients With Lower Extremity Peripheral Artery Disease: Executive Summary: A Report of the American College of Cardiology/American Heart Association Task Force on Clinical Practice Guidelines. Circulation 2017, 135, e686-e725, doi:10.1161/CIR.0000000000000470.

Treat-Jacobson, D.; McDermott, M.M.; Bronas, U.G.; Campia, U.; Collins, T.C.; Criqui, M.H.; Gardner, A.W.; Hiatt, W.R.; Regensteiner, J.G.; Rich, K.; et al. Optimal Exercise Programs for Patients With Peripheral Artery Disease: A Scientific Statement From the American Heart Association. Circulation 2019, 139, e10-e33, doi:10.1161/CIR.0000000000000623.

  1. Somewhat surprising is the number of secondary endpoints (the authors suggest estimating eleven such endpoints). When choosing a large number of endpoints, problems arise in the further statistical processing of the material. Multiple endpoints increase sample size compared to a single endpoint, as sufficient power must be provided to test all hypotheses. Complicated and the calculation of the required sample size in multiple endpoints. However, the authors only calculate the required sample size for a single endpoint (maximum walking distance in the treadmill test). Then a natural question arises - is there a need to evaluate such a number of secondary endpoints if it is not possible to obtain correct statistical results?

The Reviewer #2 is right: the ELECTRO-PAD study is only powered to assess the primary outcome. For methodological, cost, and ethical considerations this is what is done when designing a clinical trial. However, even if the sample size of a clinical trial is calculated according to a primary endpoint, this doesn’t mean that the assessment of secondary endpoints would not be of interest and importance. Secondary endpoints should indeed be considered as exploratory analyses, and “hypothesis-generating” rather than “hypothesis-based” (as this is the case for the primary endpoint). Even if the statistical results that will be obtained may suffer from a lack of statistical power, beyond the objective of addressing or not a given secondary endpoint, the results obtained will be useful to design future studies adequately powered based on such exploratory results. Consequently, we believe that the secondary endpoints of the ELECTRO-PAD study are truly needed.

Please note that two additional secondary outcomes were not described in the body of the manuscript but only in File S1. One of these secondary outcomes was added in the body of the manuscript, but the other one was not added for clarity since it was related to one of the ancillary studies. Note that we reversed the numbering of these two outcomes for clarity of the presentation (See File S1).

  1. The authors plan in case of imbalance in baseline characteristics between the 2 groups, outcome data will be analyzed using linear regression models adjusting for the baseline data significantly different between the two groups. Are these linear regression models also planned for all 11 secondary points? In addition, some of these parameters will be evaluated on a smaller sample of patients (at the Rennes Clinic).

For the parameters that will be evaluated only on a smaller sample of patients (i.e., at Rennes university hospital), no adjustment using linear regression models will be performed because of the too small sample size. For instance, this will be the case for the skin microvascular function assessment, which was designed as an ancillary study.

For the other secondary outcomes, outcome data will be analysed using linear regression models adjusting for the baseline data but only if baseline data are significantly different between the two groups. Further, in case of imbalance in baseline characteristics between the 2 groups, we will proceed to a parsimonious choice of the covariables that can be considered as the strongest predictive factors that might impact the secondary outcomes, as previously emphasized (please see references below).

Considering both the inclusion/exclusion criteria and the randomisation process, we anticipate that such imbalance in baseline characteristics will be unlikely. Again, as explained above, secondary outcomes may be considered as exploratory.

References

Holmberg, M. J., & Andersen, L. W. (2022). Adjustment for Baseline Characteristics in Randomized Clinical Trials. JAMA.

Committee for Medicinal Products for Human Use. (2015). Guideline on adjustment for baseline covariates in clinical trials. London: European Medicines Agency.

  1. About the NMES protocol. Did I understand correctly that with a bilateral lesion, the duration of the NMES session for each limb will be two times less than with a unilateral lesion? What is the reason for the different duration of NMES per day in different weeks of the program (2 hours or 3 hours)?

The Reviewer #2 is perfectly right. The participants with bilateral lesions apply NEMS on one calf during half of the program session, then on the other calf for the other half session. Thus, the duration of the NMES session for each limb for participants with bilateral lesions will be two times less than the session duration for participants with a unilateral lesion. The device used doesn’t allow the user to connect simultaneously two pairs of electrodes (i.e., one for each limb). Another possible solution would have been to use two devices simultaneously in participants with bilateral lesions; we did not choose this solution because we anticipated this would have been too burdensome for those participants, increasing the risk of low adherence to the NEMS program or study drop out.

At the time of the conception of the ELECTRO-PAD protocol, we hesitated to focus only on participants with PAD and with pain mainly located at the calves ‘level, with the presence of most symptomatic and limiting side. We finally aimed to include participants also with bilateral lesions for two reasons: i) increase the applicability of the intervention in people with PAD, if efficient; ii) do not increase the duration of the inclusion period, and the length of the study, which would have been the case by focusing only on participants with unilateral lesions.

We would like to remind that only patients with bilateral lesions leading to a walking limitation due to symptoms in both legs equally must apply NEMS in both calves. Those patients with bilateral lesions but with a walking limitation that is mainly due to symptoms in one calf, apply NEMS in the most symptomatic calf, as the participants with unilateral lesion do. We predict that there will have few participants with walking limitation due to strictly equal bilateral symptoms. When we’ll analyse the results, the exact description of the symptoms of the participants will be available. Thus, if needed, it will be possible to assess whether the change in maximal walking distance differs according to the participants’ symptoms profile (bilateral vs unilateral).

As underlined by the Reviewer #2, we start the first two weeks of the NEMS program with a NEMS duration of 3h/day (for 5 days/week). We aim to have an important dose of NEMS from the outset of the intervention hoping to maximize the potential effect of NEMS on walking capacity, if any. Then on weeks on 3 to 6, we decrease the NEMS duration to 2h/day, aware that we had to be careful on adherence of the participants to the NEMS program. Then, we again increase the NEMS duration to 3h/day during weeks 7 and 8 to increase again the dose of NEMS. We thought that these changes in the dose of NEMS would be beneficial to boost potential physiological adaptations.

  1. Patients with PAD are characterized by low motivation to participate in physical rehabilitation programs. How do the authors of the study plan to solve this problem? Wouldn't it turn out that the NMES group will have the most motivated patients for rehabilitation, which will lead to a bias in the results?

We agree with the Reviewer #2 that patients with PAD are reluctant to participate in physical rehabilitation programs. An important reason to explain this low uptake to physical rehabilitation programs relies to the fact that PAD patients must face walking pain when engaging in a walking program. However, in the ELECTRO-PAD study, we anticipate that this will probably be much less the case since a NEMS session leads to no pain while increasing lower limb blood inflow (Abraham et al., 2013).

Further, since the participants are randomized after the inclusion visit (visit #1), and since they know their group allocation only after the 7-day measurement period of physical activity, it is unlikely that the most motivated participants are more allocated in the NEMS group than in the control group.

However, we agree with Reviewer #2 that it cannot be excluded that those PAD participants who accept to participate to the ELECTRO-PAD trial are more motivated than those who declined. We do not assess participants’ motivation at the screening stage, but we collect the reasons of non-participation as far as possible. This is a common issue when supervised exercise programmes are proposed to PAD participants (Harwood et al., 2016). Finally, this is also possible that more patients in the control group will drop out the study because a deception not receiving the treatment. Since we collect the reasons in case of study drop out or study discontinuation, we will use this information to interpret our results.

References

Abraham, P., Mateus, V., Bieuzen, F., Ouedraogo, N., Cisse, F., & Leftheriotis, G. (2013). Calf muscle stimulation with the Veinoplus device results in a significant increase in lower limb inflow without generating limb ischemia or pain in patients with peripheral artery disease. Journal of vascular surgery, 57(3), 714–719. https://doi.org/10.1016/j.jvs.2012.08.117

Harwood, A. E., Smith, G. E., Cayton, T., Broadbent, E., & Chetter, I. C. (2016). A Systematic Review of the Uptake and Adherence Rates to Supervised Exercise Programs in Patients with Intermittent Claudication. Annals of vascular surgery, 34, 280–289. https://doi.org/10.1016/j.avsg.2016.02.009

Reviewer 3 Report

This study protocol focuses on the possible effect of calf neuromuscular electrical stimulation program on walking performance in PAD patients.

The introduction part is well-written and provides sufficient data about the scientific backgroun of the study.

The methods part needs some improvement as the proposed number of patients in not mentioned. Please provide some information about the calculated numer of the study group as well as the number of control patiens and details about the calculation method. Otherwise the exclusion and inclusion criteria and the description of outcomes are clear. 

The discussion part also conatins relevant information and expectations.

The advantage of the study could be the possible improving effect of electic calpf stimulation on several outcomes of PAD as comparing to exercise prorams this method is relatively cheap and reliable. 

Author Response

We thank the Reviewer #3 for his/her reviewing.

The number of participants together with details regarding sample size calculation were already described in the original version of the manuscript: subsection “2.7 Sample size” of the section 2. Materials and Methods.

We sincerely hope this will give to the Reviewer #3 the expected information.

Reviewer 4 Report

Dear Authors, I have read your manuscript with interest.

The current manuscript titled: "A multicenter, investigator-blinded, randomized controlled trial to assess the efficacy of calf neuromuscular electrical stimulation program on walking performance in peripheral artery disease: The ELECTRO-PAD study protocol" represents an important analysis of evolving field of Cardiology.

The title reflects the manuscript content and helps the reader navigate the article essence.

In my opinion, these are the adjustments which should be made to increase the value of your manuscript:

1.      In Introduction chapter, please, add more detailed information about PAD epidemiology data.

2.      Please, specify intervals of ABI values - normal and pathological.

3.      Please, change the phrase “the absence of a sufficient number of high-quality studies” in a more academic style without downplaying other research.

4.      In the Discussion section, there is not enough comparative information with other studies.

5.      Please, add supposed study limitations.

6.      It is recommended to transfer some of the information from the Supplementary Materials to the manuscript text.

7.      The manuscript contains some punctuation errors, please revise the text.

Author Response

We thank the Reviewer #4 for his/her interest in our manuscript and his/her relevant comments.

  1. In Introduction chapter, please, add more detailed information about PAD epidemiology data.

According to the Reviewer #4’ request, more detailed information about PAD epidemiology data has been added:

Lower extremity peripheral artery disease (PAD) is a serious public health concern with estimated prevalence in people aged 25 years and older of 5,5 % worldwide (236,62 million people), 7,99 % in the European Region (51,1 million people) and 9,79 % in France (4,42 million people) [1].

  1. Please, specify intervals of ABI values - normal and pathological.

This information has been added when describing the inclusion and exclusion criteria in the new Table 1 (according to comment n°6, below), as follows:

PAD is defined by an ankle–brachial index (ABI; the ratio of the systolic blood pressure at the ankle to the systolic blood pressure in the arm) of 0.90 or less. An ABI is considered as “borderline” between 0.91 and 0.99, “normal” between 1.00 and 1.40, or “noncompressible” when >1.40 [3].

  1. Please, change the phrase “the absence of a sufficient number of high-quality studies” in a more academic style without downplaying other research.

The sentence was rephrased as follows:

Owing to the very low number of available studies with adequate sample size and low risk of bias, a recent systematic review concluded that no clear clinical indication could be drawn regarding the efficacy of NMES for the management of impaired walking function in patients with PAD [13].

  1. In the Discussion section, there is not enough comparative information with other studies.

In addition to the comparative information already underlined in the Introduction section, additional comparative information with other studies has been added in the Discussion section:

For instance, the two only available randomized controlled trials testing home-based NMES programs alone to improve walking performance used transcutaneous electrical nerve stimulation as a placebo [12,14], whereas this type of lower extremity electrical stimulation was reported to increase walking capacity in people PAD in another study [53].”

Interestingly, Babber et al. recently showed a significant adjunctive benefit of a foot-plate NMES program in PFWD improvement but not in MWD improvement when NEMS was used as an adjunct to SEP as compared to SEP alone [49].”

  1. Please, add supposed study limitations.

In the original version of the manuscript, we already aimed to highlight potential study limitations when discussing about the choice of the control group (absence of sham, absence of a SEP group). It is likely that this wasn’t clearly enough outlined.

Thus, we reorganized the Discussion section by outlining first the strengths of the study protocol, and then its potential limitations. According to the Reviewer #2’ comment, we added another potential limitation regarding the duration of the NEMS sessions for each limb for participants with bilateral lesions.

We hope these modifications will give satisfaction to the Reviewer #4.

  1. It is recommended to transfer some of the information from the Supplementary Materials to the manuscript text.

The supplementary Materials contain a lot of information, and we wonder to which information in particular the Reviewer #4 is referring to.

Nonetheless, in line with the Reviewer #4’ comment, we transferred the inclusion and exclusion criteria to the manuscript text, within a Table (new Table 1). Please note that to avoid any confusion with the French terminology the term “non-inclusion” criteria was replaced by the term “exclusion” criteria.

We hope this will comply with the Reviewer #4’ request. If not, we would be grateful to the Reviewer #4 to outline which information he/she would like we transfer from the Supplementary Materials to the manuscript text.

  1. The manuscript contains some punctuation errors, please revise the text.

We apologize to the reviewer #4 for those punctuation errors.

A careful revision of the text has been done with correction of the punctuation errors when identified.

Round 2

Reviewer 2 Report

I am impressed by the great work of the authors in correcting the manuscript and satisfied with the answers to my questions. I have no other comments.

Reviewer 4 Report

I agree with the changes made, which significantly improve the quality of the manuscript. Good luck!